# Health Benefits of Cereal Grain- and Pulse-Derived Proteins

**DOI:** 10.3390/molecules27123746

**Published:** 2022-06-10

**Authors:** Jenny Bouchard, Maneka Malalgoda, Joanne Storsley, Lovemore Malunga, Thomas Netticadan, Sijo Joseph Thandapilly

**Affiliations:** 1Richardson Center for Functional Foods and Nutraceuticals, Winnipeg, MB R3T 2N2, Canada; jenny.bouchard2@agr.gc.ca (J.B.); storsley@me.com (J.S.); lovemore.malunga@agr.gc.ca (L.M.); 2Department of Food and Human Nutritional Sciences, University of Manitoba, Winnipeg, MB R3T 2N2, Canada; maneka.malalgoda@umanitoba.ca; 3Morden Research and Development Centre, Agriculture and Agri-Food Canada, Morden, MB R6M 1Y5, Canada; 4Canadian Centre for Agri-Food Research in Health and Medicine, Winnipeg, MB R2H 2A6, Canada; 5Department of Physiology and Pathophysiology, University of Manitoba, Winnipeg, MB R3E 0J9, Canada

**Keywords:** plant protein, cereal grains, pulses, bioactive peptides, chronic disease

## Abstract

Pulses and whole grains are considered staple foods that provide a significant amount of calories, fibre and protein, making them key food sources in a nutritionally balanced diet. Additionally, pulses and whole grains contain many bioactive compounds such as dietary fibre, resistant starch, phenolic compounds and mono- and polyunsaturated fatty acids that are known to combat chronic disease. Notably, recent research has demonstrated that protein derived from pulse and whole grain sources contains bioactive peptides that also possess disease-fighting properties. Mechanisms of action include inhibition or alteration of enzyme activities, vasodilatation, modulation of lipid metabolism and gut microbiome and oxidative stress reduction. Consumer demand for plant-based proteins has skyrocketed primarily based on the perceived health benefits and lower carbon footprint of consuming foods from plant sources versus animal. Therefore, more research should be invested in discovering the health-promoting effects that pulse and whole grain proteins have to offer.

## 1. Introduction

It has been established that dietary food patterns play an important role in our health and wellbeing. Consumption of fruits, vegetables, whole grains and plant-based protein sources are proven to provide long-term health benefits [1,2,3,4]. Cereal grains and pulses are primary staple foods around the world, contributing to more than half of all calories and a significant portion of protein intake. A large body of evidence demonstrates that regular consumption of whole grains and pulses confers protective effects against various chronic diseases including obesity [5,6], type 2 diabetes (T2D) [7,8], cardiovascular disease (CVD) [9,10] and cancer [11,12].

Cereal grains belong to the *Gramineae* (grasses) family that includes wheat, barley, oats, maize, rice, rye, millets and sorghum among others. Cereal grains contain approximately 70–72% carbohydrates, 7–15% protein and 1–12% lipids [12,13]. They provide a significant source of energy in most diets and may be consumed as whole grains or in their refined form. Whole grains are made up of three parts, namely the endosperm, the bran and the germ, present in relatively the same ratio as in the kernel. The endosperm accounts for 60–85% of the grain weight, making it the largest morphological component of the grain [14]. The endosperm consists of two tissues: the starchy endosperm and the aleurone layer. The starchy endosperm cells are large and irregular, have thin cell walls and contain starch granules which are embedded in a matrix of storage protein bodies and provide energy for the embryo [15,16]. The aleurone layer cells are characterized by their cuboidal shape, prominent nucleus and thick cell walls which surround the endosperm and the germ [15,16]. Aleurone cells are relatively rich in proteins, lipids, B vitamins and iron. In most cereals, the aleurone tissue is made up of a single layer of cells, but in barley, there can be as many as three layers [16]. During germination, the aleurone tissue secretes digestive enzymes that hydrolyze storage reserves in the starchy endosome to provide nutrition for the embryo [16]. The bran is the outer coating of the grain that includes the pericarp and the seed coat [17]. It accounts for 5–15% of the grain weight, depending on the type of grain [14]. The outer layers provide a tough physical barrier to protect the seed. The bran is a very rich dietary source of soluble and insoluble fibre, protein, B vitamins and bioactive phytochemicals such as phenolics, carotenoids and phytates that give the grain antioxidant and anti-inflammatory potential [14,17]. The germ (plant embryo) is the smallest component of the grain, accounting for approximately 2–4% of the dry weight [14]. The germ has a high lipid and protein content and contains vitamin E and minerals such as potassium, magnesium and zinc [9,17]. Refined grains have a reduced nutrient content due to having the germ, the bran and the majority of the aleurone layer removed during processing such as milling (wheat), polishing (rice), pearling (barley), dehulling (oats) or decortication (sorghum).

Pulses are defined as dried edible seeds harvested from 11 types of legume crops from the *Leguminosae* family. This definition excludes crops used for oil extraction (e.g., soybeans, peanuts), crops that are harvested and consumed as a green vegetable, (e.g., hydrated green peas and green beans) and crops that are used for sowing purposes (e.g., clover seeds and alfalfa) [10,18]. The Food and Agriculture Organization (FAO) of the United Nations declared 2016 the International Year of Pulses with the goal of raising public awareness about the nutritional and health benefits of pulses and the role they play in sustainable food production as a plant-based protein source [19].

Among the hundreds of known types of pulses, the most commonly consumed worldwide are beans, dried peas, chickpeas and lentils. Generally, pulses are very rich sources of protein (18–30% dry weight) and carbohydrates (50–65% dry weight) and are low in calories and fat [20]. Mature pulse seeds have three major components, cotyledons, embryonic axis, and seed coat, which generally account for 80–90%, 1–2% and 5–15% of the dry seed weight, respectively [20]. Developing seeds have an endosperm, but once the seed reaches maturity, it is vestigial. In a mature pulse seed, cotyledons serve as major nutrient reserves for early seedling growth [20]. They contain high amounts of protein, soluble fibre, resistant starch and oligosaccharides [21]. The embryonic axis contains the epicotyl (shoot), the hypocotyl and the radicle (root) [22]. It is rich in several nutrients, including B vitamins, vitamin E, phosphorous, zinc and magnesium [21]. The seed coat is a protective barrier for the cotyledons and is a concentrated source of insoluble fibre and phenolic compounds (including tannins, phenolic acids and flavonoids). Both cereal grains and pulses contain compounds such as phytic acid, tannins, lectins, enzyme inhibitors, saponins and alkaloids that are commonly referred to as antinutritional factors (ANFs) [21,23]. However, many studies indicate that the undesirable effects are linked to high doses of ANFs, and in small quantities, they may play a role in disease prevention and are beneficial for human health [24,25,26,27]. As most ANF are concentrated in the hulls, dehulling allows for their utilization for nutraceuticals, while the cotyledons can be used as a plant protein source. In addition, most deleterious effects of ANFs such as reduced mineral and amino acid digestibility can be neutralized by processing methods like soaking and heat processing.

Generally, proteins are folded in a specific biologically functional conformation strongly related to their unique amino acid composition. Additionally, proteins often occur in storage matrices called protein bodies or are physically entrapped in structures such as cell walls. This, along with their tightly folded conformation, often restricts the access of the hydrolases to their substrate which affects the digestibility of the protein and the bioavailability of the amino acids [28]. Raw plant protein sources tend to be less digestible than animal proteins due to high concentrations of insoluble fibre and the presence of ANFs. However, plant protein digestibility is significantly improved after processing, such as germination and heat treatments, including cooking [29].

Quality protein sources, which contain a balanced amino acid profile, are essential for a healthy and balanced diet. Adequate protein consumption is often a concern in diets that do not rely on animal protein. In North America, plant-based foods occupy just under 10% of the market share for protein-rich food products, though this number is projected to continually increase due to the increase in the number of consumers following flexitarian, vegetarian and vegan diets [30]. In particular, the 2020 Canada’s food guide has put an emphasis on consuming plant protein foods [31]. Recently, food proteins have been shown to be more than just sources of energy and essential amino acids. They also possess functional qualities that allow them to act as carriers for a large range of bioactive compounds such as fatty acid-rich oils, carotenoid pigments, vitamins, flavonoids and probiotic bacteria [32]. In addition, food proteins are sources of bioactive peptides [33] (Table 1 and Table 2). These peptide sequences are inactive while kept within the parent protein, and once released by methods such as enzymatic hydrolysis, microbial fermentation and gastrointestinal or chemical digestion, they possess biological activity that has a positive effect on bodily functions, human health and disease prevention [34,35].

Efforts are being made to discover the health benefits associated with an increased consumption of plant protein sources. In particular, the consumption of both cereal grains and pulses may provide a potential synergistic effect due to their nutritional and phytochemical composition as well as their complementary amino acid profiles. Recently, methods to improve the efficacy of plant protein extraction have been explored [68,69]. Furthermore, interest has shifted to include not only pulses, but whole grains such as wheat, oats and barley as important sources of plant protein. However, the literature on proteins from pulse and whole grain sources is limited compared to the literature on other protein sources like meat, milk and soy. In addition, in this review, we emphasized the importance of bioactive peptides in providing health benefits such as modulation of gut health as well as antiproliferative, anti-inflammatory, cardioprotective, hypoglycemic and hypocholesterolemic activities. In this article, the current evidence that demonstrates the importance of bioactive peptides derived from whole-grain cereal and pulse sources to the maintenance of human health and attenuation of chronic disease is reviewed in detail.

## 2. Protein Content and Amino Acid Composition

Traditionally, seed proteins are classified into four categories based on their solubility according to Osborne’s protein fractionation method [70]. Albumins are soluble in water. Globulins are soluble in salt solutions. Prolamins are soluble in alcoholic solutions and are rich in proline and glutamine. Glutelins are soluble in dilute acid or alkaline solutions with the presence of reducing or chaotropic agents and contain higher concentrations of methionine and cysteine [71]. Based on functionality, seed proteins can fall into three groups: seed storage proteins (SSPs), metabolic proteins and structural proteins which include chromosomal, ribosomal and membrane proteins [72]. SSPs account for most of the protein found in pulses and cereal grains. These proteins are synthesized in either the endosperm or the cotyledon and are compartmentalized in membrane-bound organelles called protein bodies during seed development. Generally, these proteins do not possess catalytic activity. Their main function is to store nitrogen, sulfur and carbon for germinating seedlings [73]. Most proteins from the albumin fraction have metabolic functions and consist of enzymes, protease and amylase inhibitors and lectins, while globulins, prolamins and glutelins are referred to as SSPs [71]. SSPs are further classified by their molecular weight, disulfide linkages and amino acid composition.

In pulses, globulins account for approximately 70% of the total protein. The major globulins found in pulses are legumins (11S), vicilins (7S) and, in smaller amounts, convicilins. In other types of seeds, the corresponding proteins are often classified as legumin-like or vicilin-like globulins [74]; 11S legumins have hexameric quaternary structures that contain both acidic and basic subunits, while 7S vicilins have a trimeric structure [71]. Convicilins are highly homologous to vicilins but possess an extended N-terminus that is highly charged with acidic residues and contain few hydrophobic residues. In addition, convicilins contain sulfur amino acids that are absent in vicilins [75]. Albumins represent about 10–20% of the pulse protein and contain various enzymes and proteinous ANC such as lectins and enzyme inhibitors that are thought to be defensive mechanisms for the plant. Prolamins and glutelins are present in minor concentrations in pulses [71].

In almost all cereal grains, prolamins are the most abundant class of SSPs, followed by glutelins. Rice and oats are exceptions, as they contain higher concentrations of glutelins and globulins, respectively. Albumins and globulins are concentrated in the aleurone layer, the bran and the germ, while prolamins and glutelins are located only in the starchy endosperm. The prolamin fractions of the different cereal grains are called gliadin (wheat), hordein (barley), secalin (rye), avenin (oats), oryzin (rice), zein (corn) and karfirin (millet and sorghum). Glutelin fractions have been termed glutenin (wheat), secalinin (rye), hordenin (barley) and zeanin (corn) [76]. While proteins are distributed over the whole grain, the amino acid content in cereal grains is largely determined by the amount present in the starchy endosperm, which accounts for the majority of the dry weight of the seed.

It is well-known that plant protein foods can be deficient in one or more essential amino acids when being consumed as a sole protein source. Most cereals are typically rich in glutamine, proline, leucine, methionine and cysteine while limited in essential amino acids lysine, tryptophan, threonine and valine [77]. High prolamin content is responsible for low essential amino acid content. Rice and oats contain a lower concentration of prolamins and therefore have a better balance of essential amino acids [77]. Pulse proteins are complementary to cereal grain proteins, as they generally are rich in lysine, leucine, aspartic acid, glutamic acid and arginine but are deficient in methionine and cysteine [71]. The food industry is exploring strategies to use complementary plant proteins to formulate a complete protein source while developing products [78]. For example, incorporating pulse flour into a variety of products like pastas and bakery items can improve the protein quality of these items [79,80]. Consuming pulses and whole grains in combination (such as rice and beans) can provide adequate amounts of essential amino acids and meet human protein needs. In addition, efforts are being made by breeders and plant geneticists to improve the essential amino acid content of seed proteins. As a result, high-lysine barley, maize and sorghum mutations have been identified [76,81].

The total protein and amino acid intake differ when consuming plant protein versus animal protein. However, when calorie requirements are met, plant-based diets can meet or exceed protein recommendations as plant foods contain the 20 amino acids humans require, including all the nine indispensable amino acids [82]. Evaluating the dietary protein and amino acid digestibility in foods is necessary to access the quality of the protein. The digestible indispensable amino acid score (DIAAS) is a method commonly used to assess protein nutritional value and quality. The DIAAS method is based on the amino acid digestibility of each amino acid from samples taken from the distal ileum [83]. If the values cannot be determined in humans, pigs are deemed an appropriate model. This allows for the calculation of the protein value of not only individual ingredients, but of mixed meals containing many proteins [83]. Table 3 summarizes the DIAAS score for adults in cereal grains and pulses available in the literature.

### Challenges and Future Trends

Though it is evident that bioactive peptides can play a role in ameliorating chronic disease risk factors, the development of peptide-based products can present certain challenges which may impact the bioavailability, consumer acceptability and commercialization of these products. First, peptides containing hydrophobic residues such as proline and phenylalanine can have a bitter taste [93]. This can impact the overall taste of the product and negatively affect consumer acceptability. Debittering peptides and protein hydrolysates can be achieved by several methods reviewed by Fitzgerald and O’Cuinn [94]. However, this may come at the expense of the bioactivity of the peptides. Second, the metabolic stability and bioavailability of the peptides is an incredibly important factor to consider. The molecular size, charge and hydrophobicity of peptides influence their bioavailability [95]. When orally consumed, peptide bonds are susceptible to gastric proteases which may lead to a loss of the structure of the peptide and, consequently, of its intended function [93]. The discovery of bioactive peptides that are not susceptible to cleavage by digestive enzymes is an area of research worth focusing on. Technologies such as microencapsulation of peptides may overcome this challenge, though this can substantially increase the manufacturing costs and impact the affordability of the resulting products [93]. Third, there is an abundance of in vitro studies available in the literature; however, there is a lack of preclinical and clinical trials to substantiate the efficacy of the bioactive peptides derived from plant sources in vivo. Emphasis should be placed on providing clinical evidence to support any future health claims on products [93].

## 3. Evidence of Health Benefits of Cereal and Pulse Grain Proteins

A large body of research has evidenced that plant protein is able to attenuate risk factors for chronic disease. In the following sections, the health benefits of cereal grain and pulse proteins are discussed in more detail. Figure 1 summarizes the potential mechanisms in which pulse and whole grain proteins exert beneficial health effects.

### 3.1. Obesity

Obesity is a chronic disease defined as an abnormal or excessive body fat accumulation. Western diets containing high intakes of red processed meat, salt, simple sugars and saturated and trans fats are often associated with higher incidence rates of obesity [96]. Obesity has been established as a risk factor for developing other chronic diseases including T2D, CVD, osteoarthritis, certain cancers and neurodegenerative diseases such as Alzheimer’s disease [97]. Increased consumption of cereal and pulse grain proteins has been shown to increase satiety and can help with weight management. The incorporation of pulses in meat products such as sausages and burgers and baked goods in the form of flour could improve the fibre content, texture and sensory properties of the products as well as reduce their caloric content and carbon footprint [98,99].

It has been reported that both pea and wheat proteins significantly increased the release of appetite-modulating hormones cholecystokinin (CCK) and glucagon-like peptide 1 (GLP-1) in human duodenal tissue [100]. Abete et al. (2009) conducted a study that consisted of groups of obese human subjects consuming various hypocaloric diets, including a legume-rich diet, for 8 weeks. The results showed the legume-rich diet increased mitochondrial oxidation, which contributed to a greater weight loss compared with the controls, similarly to the group eating a high-animal protein diet. In addition, the legume-rich diet also lowered the total and LDL-cholesterol and was the only dietary approach in the study that lowered blood pressure [101]. Most pulse and cereal grains are relatively rich in glutamine, which has been shown to increase postprandial energy expenditure by 49% due to elevated carbohydrate and fat oxidation in human subjects [102]. Pulse lectins and enzyme inhibitors have been proposed as therapeutic agents for preventing/controlling obesity. Red kidney bean phytohemagglutinin (PHA) reduced insulin secretion levels and fat deposition in obese Zucker rats without loss of muscle and body protein [103] while inducing CCK secretion from duodenal mucosa and stimulating pancreatic growth in Sprague Dawley and Lister hooded rats [104]. A *Phaseolus vulgaris* extract called Beanblock (containing PHA and an α-amylase inhibitor) has been reported to reduce postprandial glucose and insulin levels as well as suppress ghrelin production, inducing less of a desire to eat in healthy human subjects [105]. In addition, several clinical studies have demonstrated that administering a Phase 2 odorless and tasteless proprietary α-amylase inhibitor produced from non-GMO white kidney beans facilitated significant weight loss in overweight/obese individuals [106,107].

Peptides derived from rice protein may possess anti-adiposity potential. An animal study conducted by Yang et al. demonstrated that rice protein was able to improve body weight and adiposity by upregulating lipolysis and downregulating lipogenesis which resulted in lower serum and hepatic lipid levels in rats [108]. Another animal study by Kannan et al. isolated a pentapeptide from rice bran that possessed protective effects against obesity in preadipocytes and Alzheimer’s disease in neuronal cells. In addition, rice protein suppressed the growth of *Proteobacteria* in mice, which would reduce endotoxin production and reduce inflammation associated with insulin resistance [52]. This suggests that rice protein may help maintain gut microbiota diversity and protect against obesity and other related diseases [109].

Though there is plenty of evidence of the beneficial effects of consuming cereal grain fibre, there are fewer studies on the effects of protein from whole grain sources on weight loss and weight management in humans. As plant proteins are gaining popularity in the marketplace, more research is needed to establish the anti-obesity effects of whole grain-derived proteins.

### 3.2. Hypercholesterolemia

Hypercholesterolemia is defined as having abnormally elevated blood concentrations of LDL-C and/or non-HDL-C and is a well-documented modifiable risk factor for CVD [110]. Combined results from Cycles 5 (2016–2017) and 6 (2018–2019) of the Canadian Health Measures Survey determined that 28% of Canadians aged 18–79 have hypercholesterolemia. The prevalence increases significantly with age, as this number surges to 60% in Canadians aged 60–79 [110]. Hereditary factors often play a role, but a diet that is high in saturated and trans fats increases the rate in which the liver synthesizes LDL- and VLDL-C.

Pulses and whole grains contain an abundance of bioactive compounds other than proteins, such as dietary fibre, resistant starch, mono- and polyunsaturated fats, tocols, isoflavones and plant sterols, which have been shown to beneficially alter lipid metabolism [111]. Health Canada has substantiated two health claims acknowledging a link between the daily consumption of 3 g of β-glucan, a soluble fibre found in oats and barley, and the reduction of serum LDL-C and TC which effectively reduces the risk of heart disease [112,113].

It has been established that dietary proteins influence lipid metabolism. Based on in vitro and in vivo studies, the cholesterol-lowering mechanisms of peptides found in food involve (a) the binding to bile acids derived from cholesterol to form insoluble components that prevent their reabsorption, (b) disrupting cholesterol’s micellar solubility formed by bile salts, (c) altering expression and activity of hepatic enzymes related to cholesterol synthesis and metabolism and (d) favorably modulating the gut flora (Figure 2) [114,115].

Chickpea and lentil proteins are reported to effectively lower plasma VLDL-C and plasma and liver triglyceride (TG) concentrations as well as decrease lipoprotein lipase activity in epididymal fat and increase hepatic lipase activity in Wistar rats compared to the casein control group [116]. A study by Shi et al. revealed that chickpea peptides were effective in decreasing the total serum TGs and the total and LDL-cholesterol, increasing HDL-C levels and inhibiting fatty acid synthase and 3-hydroxy-3-methyl-glutaryl-coenzyme A reductase (HMGR) activities in high-fat diet-induced obese rats [65]. In addition, the authors identified a peptide, VFVRN, that was found to effectively inhibit HMGR in a pharmacore model, as well as inhibit TC synthesis by reducing expressions of HMGR, sterol regulatory element-binding protein-1c and -2 (SREBP) and liver X receptor in HepG2 cell lines. HMGR is a crucial rate-limiting enzyme in the mevalonate pathway to synthesize cholesterol [65]. Furthermore, they determined that chickpea proteins reduce cholesterol micellar solubility in vitro. Cholesterol is a water-insoluble molecule that requires micelle formation to be absorbed by the intestines. Disrupting micellar solubilization allows for increased fecal excretion of cholesterol and bile acids, thus contributing to cholesterol-lowering effects in animals and humans [65]. Lentil protein hydrolysate (LPH) was observed to effectively bind to bile acids in vitro [117]. In addition, LPH was found to increase plasma TC and HDL-C levels but decrease the atherogenic index in obese Zucker rats. Fat content in the liver was reduced and the hepatic fatty acid profile was altered via significant reduction of saturated fatty acids and a significant increase in monounsaturated fatty acids. LPH administration also upregulated the hepatic mRNA expression of SREBF-1 and Fasn, which are genes involved in lipid metabolism [118]. Yahia et al. reported that chickpea protein hydrolysate (CPH) significantly decreased the serum and liver TC and increased the HDL:VLDL + LDL-cholesterol ratio in hypercholesterolemic rats. In addition, CPH was able to increase the activity of lecithin–cholesterol acyltransferase, a key liver enzyme responsible for the conversion of unesterified cholesterol to cholesterol ester despite low serum cholesterol and HDL-C levels, allowing for increased cholesterol reverse transport efficiency [119]. Oxidative stress is a factor that links hypercholesterolemia with the pathogenesis of atherosclerosis via the overproduction of free radicals that lead to lipid peroxidation and an inadequate antioxidant defense system. CPH was also shown to potently decrease lipid peroxidation and improve serum malondialdehyde and hydroperoxide levels [119]. An in vitro study concluded that CPH had the DPPH radical scavenging activity that was 2.16-fold more effective than the chickpea isolate, indicating the degree of hydrolysis has a significant effect on the bioactivity of these peptides [55]. Amaral et al. compared the effects of a daily dose of chickpea legumin, 11S globulin, to the drug simvastatin administered to hypercholesterolemic rats. The results showed that chickpea legumin had no effect on serum cholesterol concentration, but was shown to decrease serum TGs by 28.77 and 23.53% compared to the control and simvastatin groups, respectively. This suggests that the 11s fraction may exert an effect on TG metabolism independent of changes in cholesterol concentrations [120]. Human clinical trials are needed to further assess the hypocholesterolemic properties of chickpea protein.

Lupins are yellow legumes that are popular in areas that have a drier climate or high-saline conditions which are not ideal for soybean cultivation such as the Mediterranean region or Oceania. Lupin has gained significant interest due to various nutritional and health benefits [121]. Clinical trials have confirmed that lupin protein supplementation can effectively reduce the LDL-C/HDL-C atherogenic index and TC levels, with the most pronounced effects in patients with high baseline LDL-C/HDL-C levels [122,123,124]. Another clinical trial demonstrated that lupin protein supplementation reduced TC, LDL-C and non-HDL-C concentrations as well as a significantly decreased PCSK9 levels in moderately dyslipidemic patients [125]. PCSK9 is considered a pro-artherogenic molecule as it binds to LDL receptors in the liver, prevents their recycling and promotes their lysosomal degradation, which results in reduced cellular uptake of LDL particles and an elevation in LDL-C levels [126]. In vitro studies have shown that peptides derived from lupin protein hydrolysate such as P3 (YDFYPSSTKDQQS), P5 (LILPKHSDAD) and P7 (LTFPGSAED) are effective HMGR inhibitors. P3 is able to increase activation of the SREBP-1 pathway to improve LDL receptor protein levels [127] while P5 was found to also inhibit PCSK9, both in HepG2 cells [66].

Cereal grains have also been reported to be effective against hypercholesterolemia. Rice protein has been proven effective in modifying TG metabolism and improving lipid homeostasis. Rice protein is thought to decrease the hepatic secretion of TGs and cholesterol by upregulating lipolysis, downregulating lipogenesis and interfering with very-low-density lipoprotein (VLDL) synthesis and secretion, lowering lipid accumulation, as shown in normal Wistar rats [128]. Rice protein was also shown to increase the activity of mRNA levels of cholesterol 7α hydroxylase and decrease the activity and gene expression of acyl-CoA cholesterol acyltransferase in hypercholesterolemic rats [108]. Ronis et al. reported that the consumption of RPI offered protective effects against problems associated with a high-fat Western diet such as insulin resistance, hypercholesterolemia and steatosis in Sprague Dawley rats. The authors also found RPI to inhibit expression of hepatic genes involved in fatty acid synthesis [129]. Another study determined that rice protein was able to lower serum and hepatic TC as well as hepatic total lipid levels in Sprague Dawley rats by increasing fecal TC and bile acid excretion [130]. Rice protein was shown to increase antioxidative capacity in normal adult male Wistar rats by increasing activities of superoxide dismutase and catalase, as well as stimulation of glutathione synthesis. This resulted in significantly reduced plasma TC levels. As previously mentioned, oxidative stress is an important factor in hypercholesterolemia. Therefore, the increased antioxidative response should prevent oxidative damage to lipids and proteins, contributing to the lipid lowering effects exerted by rice protein [131]. Additionally, a single clinical trial reported that the administration of a daily dose of 10 g of rice endosperm protein for four weeks was able to increase HDL-C levels and reduce serum uric acid levels in adult male subjects with metabolic syndrome risk factors. However, no changes in TGs or LDL-C were observed [132]. Additional clinical trials are needed to evaluate the antioxidative and hypocholesterolemic effects of rice protein.

Limited research has shown that wheat gluten has the potential to play an important role in cholesterol metabolism. Liang et al. reported that wheat gluten supplementation significantly reduced serum TC and LDL-C concentrations as well as liver TC, FC, CE and TG concentrations in hypercholesterolemic hamsters. In addition, increased fecal excretion of lipids, TC and bile acids was observed. Wheat gluten was found to increase short-chain fatty acid (SCFA) production which beneficially modulated the gut flora by lowering the Firmicutes to Bacteroidetes ratio (F/B ratio). A higher F/B ratio is associated with obesity, lipid metabolism and other metabolic diseases. This suggests that wheat gluten can alter intestinal microflora to regulate cholesterol metabolism [114]. In a clinical trial, increased gluten consumption was associated with a 13% reduction in serum TGs in hyperlipidemic subjects [133].

A couple of studies have evaluated the lipid-lowering effects of oat proteins. Guo et al. investigated the hypocholesterolemic effects of five oat varieties with similar β-glucan content but different protein and lipid content in hypercholesterolemic rats. The results indicated that all the five oat varieties were effectively able to increase fecal bile acid excretion while reducing plasma TC and LDL-C as well as liver TC and CE levels. In addition, the hypocholesterolemic effects were more pronounced in varieties containing higher contents of protein and lipids [134]. The same laboratory conducted a second study where three groups of hypercholesterolemic hamsters were fed different experimental diets: one containing oat protein, one containing β-glucan and one consisting of a combination of the two components. The results showed oat proteins were more effective than β-glucan in reducing plasma LDL-C and liver TC levels by increasing fecal bile acid, TC and total lipid excretion. In addition, oat protein was found to regulate liver CYP7A1 activity [135]. Furthermore, these effects were maximized in the combination group. This suggests that oat protein and β-glucan can produce a synergistic effect to increase the hypocholesterolemic effect of oats, and oat protein also exerts this effect through different physiological mechanisms than β-glucan. More animal studies and, eventually, clinical trials are needed to assess the lipid-lowering effects of oat proteins.

As evidenced by the previous section, current research concludes that not only the fibre, but the protein found in pulses and whole grains possesses lipid-lowering properties that may act in synergy with other bioactive compounds to attenuate markers of hypercholesterolemia.

### 3.3. Diabetes

Diabetes mellitus is a group of metabolic disorders that are characterized by high blood glucose levels over prolonged periods of time resulting from insulin resistance, inability to produce insulin or both. If untreated, diabetes can lead to many life-threatening health complications including diabetic ketoacidosis, microvascular problems such as neuropathy, nephropathy and retinopathy and macrovascular problems such as CVD, stroke and peripheral vascular disease.

Diabetes can be classified into four subgroups: type 1 (T1D), type 2 (T2D), gestational diabetes and diabetes associated with genetic defects and pancreatic diseases. T1D results from autoimmune destruction of pancreatic β cells that cause insufficient insulin production. T2D begins with insulin resistance and, eventually, lack of insulin as the disease progresses. Gestational diabetes is a temporary condition in which women develop high serum glucose levels during pregnancy that is usually resolved after the birth of the baby. However, women have an increased risk of developing T2D afterward. Several forms of diabetes are associated with genetic defects in β cells. This includes mutations in the glucokinase and hepatocyte nuclear factors 1 and 4 genes, genetic abnormalities that inhibit the conversion of proinsulin to insulin, endocrinopathies like Cushing’s syndrome and glucagonoma which antagonizes insulin action and excessive damage to the pancreas caused by conditions such as pancreatitis, pancreatic carcinoma and pancreatic fibrosis [136].

The prevalence of T2D is rapidly rising around the world. However, the consumption of whole grains and pulses, which have a low-medium glycemic index, is thought to reduce the risk of T2D or help manage the disease based on their fibre and phytochemical content by improving glucose metabolism and insulin sensitivity. Refined grains, on the other hand, have a high glycemic index due to the removal of fibre, protein and other valuable nutrients and may increase the risk or exacerbate symptoms [137].

Bioactive peptides derived from grains and pulse proteins may play a role in managing diabetes. Dipeptidyl peptidase IV (DPP-IV) is a serine protease that rapidly degrades incretins such as glucose-dependent insulinotropic polypeptide (GIP) and GLP-1. Extending the half-life of incretins by utilizing food-derived bioactive peptides is a promising therapeutic approach for regulating glucose homeostasis in T2D patients. DPP-IV inhibitory peptides have been identified in many plant-based sources such as rice bran [138], corn [139], oat [50] chickpea [140], lupin [141] and common bean [59,142] protein hydrolysates.

Recent studies have shown that lupin proteins have diabetes prevention potential. A study conducted by Magni et al. demonstrated that γ-conglutin, a lupin-derived protein, was able to significantly decrease postprandial glycemia in rats receiving a constant glucose intake overload in a dose-dependent manner. In addition, γ-conglutin was able to bind to insulin, primarily through electrostatic forces in vitro [143]. The effects of γ-conglutin were compared to glibenclamide, a common drug used to treat diabetes, in streptozotocin-induced diabetic rats. The results showed that rats administered γ-conglutin had higher Ins-1 expression, increased insulin levels and lower blood glucose levels than the control group, although these effects were less potent than those of glibenclamide [144]. Teruzzi et al. conducted an in vitro experiment that suggests that myocyte insulin receptors are targeted by γ-conglutin, suggesting that it possesses insulin-mimetic properties [145]. Human studies have also assessed the hypoglycemic effects of γ-conglutin. Bertoglio et al. (2011) conducted a placebo-controlled four-week trial that demonstrated that γ-conglutin administered before carbohydrate consumption exerted a hypoglycemic effect in healthy adults despite no significant variations in the observed insulin levels [146]. In addition, the addition of γ-conglutin to a sugary beverage was reported to acutely reduce glycaemia in type 2 diabetic individuals, suggesting that lupin protein could be a valuable tool in glycemic management [147].

Hypoglycemic effects have also been observed in proteins derived from whole grain sources. Rice bran protein hydrolysate was shown to decrease HOMA-IR and blood glucose and lipid values while increasing adiponectin levels and suppressing leptin levels and proinflammatory cytokine secretion in high-fat–high-carbohydrate diet-fed rats. Adiponectin is an antidiabetic adipokine that is known to increase glucose uptake and fatty acid oxidation and suppress hepatic gluconeogenesis [148]. Increased apidonectin levels due to rice protein supplementation also improved insulin resistance and was effective in preserving renal function by exerting protective effects against albuminuria and renal glomeruli damage in Goto–Kakizaki rats, a nonobese T2DM model [149]. Additionally, Ishikawa et al. concluded that rice protein hydrolysate reduced glycemic response in normal Sprague Dawley rats by stimulating GLP-1 and insulin secretion while also attenuating plasma DPP-IV activity [150]. An interesting in vitro study conducted by Guo et al. utilized a natural, edible aerogel made from sodium alginate and chitosan to protect wheat α-amylase inhibitors from simulated gastrointestinal digestion. The wheat protein-functionalized aerogel particles showed a high inhibition rate for 2 h after digestion and being transferred into intestinal juices. This innovative strategy could possibly benefit T2DM patients by safely reducing spikes in postprandial blood glucose levels [151].

Oat oligopeptides have been reported to possess antioxidant properties in vitro and inhibitory activities against α-amylase and lipase in silico, which would improve digestion of sugars and lipids [152]. Wang et al. concluded that administering 2 g/kg body weight of oat oligopeptides to diabetic Sprague Dawley rats fed a high-calorie diet resulted in significantly reduced fasting blood glucose levels and oral glucose test tolerance, as well as reduced insulin resistance and urine volume (homeostasis model assessment) [153]. A couple of clinical trials have evaluated the hypoglycemic effects of whole grain-derived protein. In a randomized crossover acute feeding trial, Tan et al. evaluated whether the addition of 24 g of oat, rice or pea protein in a sugar-sweetened beverage attenuated the glycemic response in healthy human male subjects. Insulin iAUC was significantly higher in pea and oat protein beverages, and insulin sensitivity was higher after the rice protein treatment. Furthermore, subjects reported an increase in fullness and a decrease in the perception of hunger [154]. These studies provide evidence that food-based interventions such as protein supplementation are useful in regulating glycemic control and can be helpful in managing diabetes.

### 3.4. Cardiovascular Disease

Cardiovascular disease refers to a number of conditions that involve plaque buildup in the arteries supplying blood to the heart that can lead to a heart attack, heart failure, a stroke or death. CVD is a major problem for public health and is the second leading cause of death in Canada [155]. A diet that is high in saturated fats and cholesterol is an important risk factor for CVD. Whole grains and pulses are high in fibre, micronutrients and plant protein, which are important for lowering cholesterol levels and the risk of heart disease.

Hypertension and atherosclerosis are important risk factors for CVD. Hypertension is defined by the American Heart Association as having blood pressure that is 130 mm Hg (systolic) over 80 mm Hg (diastolic) [156]. Hypertension is colloquially known as “the silent killer” as many people with hypertension are asymptomatic and do not know they have it. Left untreated, hypertension can cause organ damage to the heart, liver and brain, among others, and increase the risk of a stroke, heart attack and coronary heart disease [157]. Atherosclerosis is a slowly progressing chronic disorder that is characterized by the accumulation of atherosclerotic plaques in the arteries. It involves the accumulation of lipids in artery walls, an infiltration and inflammatory response of white blood cells such as macrophages and T cells, the formation of fibrous connective tissue by vascular smooth muscle cells and the eventual calcification of artery walls. Over time, plaque growth can cause arterial stenosis and contribute to ischemic heart disease [158].

A wealth of research has focused on the favorable inhibitory activity of bioactive peptides in cereal and pulse grain proteins. Researchers have particularly taken an interest in angiotensin I-converting enzyme (ACE) inhibitory peptides. ACE plays a key role in the renin–angiotensin system and raises blood pressure by cleaving angiotensin I to form angiotensin II, a vasoconstrictor peptide, and by catalyzing the degradation of bradykinin, a vasodilator. The inactivation of this enzyme would reduce hypertension and the risk of cardiovascular disease. While synthetic ACE inhibitors are commonly used, they are often accompanied by undesirable side effects. Therefore, identified by low cost, natural sources of ACE inhibitors have piqued interest. Researchers have analyzed many food-derived ACE inhibitor peptides from various plant sources including peas [58,159], lentils [57], chickpeas [160], common beans [59], mung beans [56,161], oats [36,37], barley [38,162], wheat [41,42], corn [39,40] and rice [43,44,45] in silico and in vitro. However, these peptides are dose-dependent, and the potency may differ in vivo due to the digestion by gastrointestinal enzymes such as pepsin, trypsin and chymotrypsin [163].

Regular consumption of high-protein, high-fibre pulses has been shown to provide cardiovascular benefits. A study by Gomes et al. demonstrated that fresh-ground common bean protein hydrolysate equivalent to a daily serving of cooked beans significantly reduced plasma TC and TG concentrations after nine weeks in atherogenic mice [164]. Porres et al. evaluated the effects of administering a lentil protein hydrolysate to obese Zucker rats with cardiac hypertrophy individually or in combination with aerobic exercise. The hydrolysate alone improved some electrocardiographic parameters, decreased plasma activity of ACE and improved the kidney function, but also provided many synergistic effects with exercise [165]. Belski et al. reported that consuming a lupin-enriched diet over a period of 12 months can lower blood pressure and blood cholesterol concentrations as well as improve insulin sensitivity in human subjects [166].

Whole cereal grains have been considered heart-healthy due to their high concentrations of fibre such as β-glucan (found in oats and barley), arabinoxylan, cellulose and pectins; however, protein from cereal grains has also been shown to have beneficial effects on the heart. A foxtail millet protein hydrolysate fed to spontaneously hypertensive rats (SHR) was effective in significantly reducing blood pressure and lowering the ACE activity and angiotensin II levels compared to the controls [167]. Rice protein hydrolysate was observed to have a protective effect on cardiomyocytes H9C2 against hydrogen peroxide-induced proliferation suppression and apoptosis [168]. Two studies have shown that a diet supplemented with rice protein isolate (RPI) was able to reduce the size of atherosclerotic lesions in apoE^−/−^ mice [169,170]. A risk factor associated with the pathogenesis of atherosclerosis is high levels of LDL-cholesterol, which results in an increase in oxidative products of LDL (oxLDL) that bind to scavenger receptors on macrophages, promoting the formation of foam cells, which contribute to the development of atherosclerotic fatty streak lesions [170]. In one study, RPI reduced the circulating levels of oxLDL and oxLDL autoantibodies despite no change in serum lipid levels and increased the expression of aortic glutathione and antioxidant enzymes. It is proposed that the mechanism by which RPI exerts atheroprotective effects is by inhibiting oxidative stress that results in the initiation and progression of atherosclerosis [170]. In addition, cell death (apoptosis and necrosis) plays a major role in several chronic diseases including CVD. Oxidative stress is known as an important proapoptotic factor. Therefore, the discovery of natural bioactive substances that attenuate oxidative stress is very useful in the fight against CVD.

Rice bran protein has been proven to possess antihypertensive effects. Shobako et al. determined that thermolysin-digested rice bran (TRB) was able to reduce systolic blood pressure with a single oral dose of 30 mg/kg in SHRs. In this study, they also identified two functional peptides, YY and LRA, which exhibited blood pressure-lowering effects at the doses of 0.5 mg/kg and 0.25 mg/kg, respectively [46]. LRA is reported to have potent vasorelaxant activity [46]. A human clinical trial conducted by the same laboratory determined that administering 1 g of TRB containing 43 µg of LRA daily for 12 weeks reduced the systolic blood pressure in subjects with high-normal blood pressure and grade 1 hypertension [171].

Fermented rice bran (FRB) has also been evaluated for heart-healthy properties. Fermentation is a cost-effective method often used to produce antihypertensive peptides derived from food proteins. FRB has been shown to possess ACE inhibitory activity in vivo. Additionally, FRB has not only exhibited antihypertensive effects in stroke-prone SHRs, but has also been able to improve glucose metabolism as well as liver TC and TG levels [172]. There is a possibility that nonpeptide compounds in FRB contribute to the antihypertensive effects. For this reason, the isolation of functional peptides in FRB and human clinical trials evaluating their effects are necessary.

### 3.5. Cancer

Cancer is characterized by the development of abnormal cells that divide in an uncontrollable manner which invade and destroy normal body tissues. These cells have the capacity to metastasize to other parts of the body to create a systemic disease [173]. Metastasis drastically increases the risk of mortality and it is speculated that it causes more than 90% of deaths related to cancer [174]. Cancer is the primary cause of mortality in Canada and is responsible for 30% of all deaths [175].

Like many chronic diseases, an unhealthy diet is an established risk factor for many types of cancers. Eating plant-based foods has been associated with a reduced risk in developing cancer. Pulses and whole grains are a rich source of fibre, polyphenols, carotenoids and sulfur compounds that likely produce anti-inflammatory and antioxidative effects that prevent the development of cancer and improve the prognosis of cancer patients [176]. In addition, recent studies evaluating the chemopreventive activity of bioactive peptides derived from pulse and whole grain sources have provided promising results.

Protease inhibitors (PI) are commonly found in many plants and are particularly abundant in legumes and whole grain seeds. PIs are important for two different roles: they are a major component of the plant’s defense mechanism against stress, insects, herbivores and microbial organisms and they are involved in regulating endogenous plant proteases [177]. Recently, PIs have been investigated as potential anti-inflammatory, antiproliferative and neuroprotective agents in humans [178]. Bowman–Birk inhibitors (BBIs) are extremely stable serine proteases with two inhibitory domains that are highly resistant to acidic conditions and other proteolytic enzymes. They are commonly known by their capacity to inhibit trypsin and chymotrypsin in the digestive tract, which has been associated with reduced bioavailability of dietary proteins [178,179]. However, recent evidence has shown that BBIs from legumes may exert chemopreventive effects. It is hypothesized that chymotrypsin-like proteases are implicated in carcinogenesis [180]. Many studies have reported that BBIs from various pulse and whole grain sources such as peas, chickpeas, lentils, kidney beans, barley and millet possess anticarcinogenic activity against several types of cancer cell lines [178,179,181]. More research and clinical trials are needed in this domain to establish BBIs from pulses and whole grains as effective chemopreventive agents.

A handful of studies have assessed the antiproliferative effects of chickpea protein. Three studies from the same laboratory evaluated chickpea peptides and lectins. The studies concluded that peptides generated from alcalase digestion induce apoptosis in a variety of human cancer cell lines through downregulation of Bcl-2 and caspase-3 activation while promoting cell cycle arrest. The chickpea lectins demonstrated anticancer potential against the same cancer cell lines while exerting DNA protection in a dose-dependent manner [182,183,184]. Another study investigated oxovanadium complexes of chickpea seed protein hydrolysate against lung cancer A549 cell lines and determined that the complexes not only exerted an inhibitory effect of cell growth, but also one that was 1.7-fold more effective than that of vanadium salt alone [185]. Chickpea protein isolate has also been reported to reduce the incidence of pre-carcinogenic legions in the colon as well as reduce LDL and TC levels in the Institute of Cancer Research mice treated with the carcinogen azoxymethane on a hypercaloric diet [186]. Cell cycle blockers are important targets for cancer treatment as abnormal cell cycle protein expression leads to cancer cell proliferation. In addition, activation of apoptotic pathways is a critical mechanism by which chemotherapies destroy cancerous cells as defects in these pathways cause the formation of tumors and their resistance to cytotoxic drugs [187].

Limited research has evaluated antiproliferative effects of common bean protein. Luna Vital et al. conducted two studies evaluating the effects of common bean protein hydrolysate on human colon cancer cells. Five peptides were shown to possess antiproliferative properties against human colon cancer cells by modifying expression of cell cycle regulation and apoptotic proteins such as p21 and cyclin B1, BAD, cytC, c-casp3, survivin, BIRC7 [60]. In the second study, the protein hydrolysate upregulated the genes that encoded the antioxidant enzymes related to NRF-2 which caused the induction of apoptosis and cell death of colon cancer cells [188].

Rice has been recognized for its beneficial effects against carcinogenicity. Rice bran, in particular, is known to contain many bioactive compounds such as gamma-oryzanol, tocotrienols, tricin and phytic acid that are effective in reducing inflammation and promoting cell cycle arrest and cell apoptosis [148]. Recently, more studies have been evaluating the effects of protein content in rice, including the bran fraction. Rice bran protein hydrolysate has been shown to inhibit growth of breast and colon cancer cell lines [189]. A pentapeptide recently isolated from rice bran, EQRPR, has demonstrated antiproliferative effects against colon, lung, breast and liver cancer lines [51]. In addition, an in silico study revealed that the same pentapeptide is able to strongly interact with integrins, which is important for anticancer effects [190]. RPI has been reported to induce apoptosis in human breast carcinoma and myeloma cell lines through induction of p21 and inhibition of CDK4 and cyclin D1 activities [191]. The prolamin fraction of rice protein is reported to stimulate an antileukemia response in human mononuclear cells [192]. In addition, Lui et al. demonstrated that rice prolamin fractions inhibited leukemia L120 cell and human leukemia Jurkat cell viability in vitro as well as decreased tumor weight and leukemia-induced liver and spleen weight reduction in L120-bearing DBA/2 mice after oral administration. In addition, RPI inhibited mammary tumor progression induced by 7,12-dimethylbenz[a]anthracene in female Sprague Dawley rats [193].

Lunasin is a 43-amino acid peptide that was originally found in soy but has since been isolated from barley, rye, oats and wheat [194,195,196,197]. It contains an RGD motif that allows for cell adhesion and internalization [198]. Lunasin is able to translocate to the nucleus of cancerous cells and inhibit acetylation of histones H3 and H4 and binds to deacetylated histones H4. This is discerned by the cell as abnormal growth and ultimately results in the activation of apoptotic pathways. Cell culture confirms that normal cell proliferation and morphology are not affected by lunasin treatment [198]. Lunasin has been reported to halt cell cycle progression at the G1/S phase through increasing p27Kip levels, inhibiting retinoblastoma phosphorylation, and modifying the expression of cyclin-dependent kinase complex components in non-small-cell lung cancer cells [199]. In addition, it promoted apoptosis and increased expression of the p21 and p27 proteins in HT-29 and KM12L4 metastatic colon cancer cells [200]. Barley lunasin has been found to suppress ras-induced colony formation in mouse fibroblast NIH3T3 cells and inhibit histone acetylation in vivo in the same cells as well as human breast MCF-7 cells in the presence of sodium butyrate, a histone deacetylase inhibitor [195]. Research indicates that lunasin is a nontoxic potential therapeutic agent against several types of cancer that may be able to combat metastasis in cases where chemotherapy resistance develops.

As technology advances, the fight to cure cancer is an ongoing battle. Conventional cytotoxic drugs are effective against cancers but are accompanied by several undesirable side effects that make the recovery of patients very difficult. The discovery of plant-derived chemopreventive peptides is becoming increasingly valuable for cancer prevention and may change the course of treatment in the future. As research continues in this domain, nontoxic bioactive compounds may soon be able to act as therapeutic agents alone or increase the efficacy of the existing chemotherapies, allowing for shortened treatment times and an easier recovery for patients after treatment.

## 4. Conclusions

Pulses and whole grains contain many bioactive compounds such as dietary fibre, resistant starch, phenolic compounds and mono- and polyunsaturated fats. Among them, recent evidence demonstrates that protein derived from pulse and whole grain sources is not only important for nourishment and sustenance, but also contains bioactive peptides that aid in the prevention of chronic diseases and risk factors such as obesity, CVD, hypercholesterolemia and diabetes. More research, especially clinical trials, is needed to evaluate the effects and elucidate the mechanisms in which pulse and whole grain proteins are able to provide disease-fighting benefits.

## Figures and Tables

**Figure 1 molecules-27-03746-f001:**
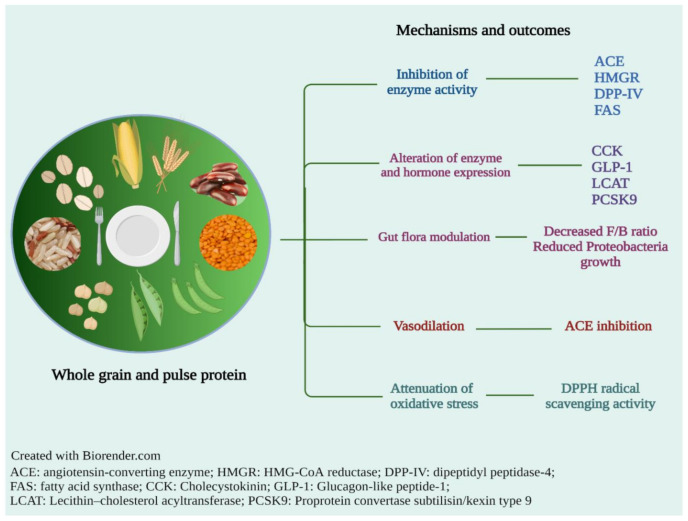
The mechanistic actions and outcomes of cereal grain- and pulse-derived proteins to promote good health.

**Figure 2 molecules-27-03746-f002:**
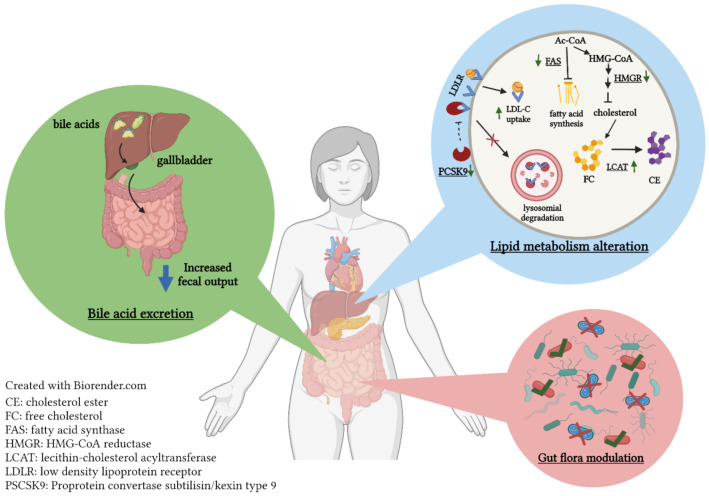
The mechanisms of action in which plant protein alters lipid metabolism.

**Table 1 molecules-27-03746-t001:** Bioactive peptide sequences from some proteins derived from whole grains.

Activity	Type of Protein	Bioactive Peptide Sequence	IC_50_	Description	Reference
Antihypertensive	Naked oat globulin	SSYYPFK	91.82 µM	ACE inhibitor; alcalase, flavourzyme, pepsin and trypsin digestion	[36]
Antihypertensive	Oat 11S and 12S globulin	GQ, QC, GL, PQ, AG	30–50 µg/mL	ACE inhibitor; thermolysin digestion	[37]
Antihypertensive	Barley protein concentrate	FQLPKF, GFPTLKIF, ALRYFM, NFLARF	28.2, 41.2, 200, 100 µM	ACE inhibitor; papain digestion	[38]
Antihypertensive	Corn gluten hydrolysate	AY	0.037 mg/mL	ACE inhibitor; antihypertensive effect in rats	[39]
Antihypertensive	Corn gluten hydrolysate	PSGQYY	100 µM	ACE inhibitor; antihypertensive effect in rats; pescalase digestion	[40]
Antihypertensive; anticancer	Wheat germ protein hydrolysate	SGGSYADELVSTAK, MDATALHYENQK	0.09, 0.21 µM	ACE inhibitor; inhibited A549 lung cancer cell growth; proteinase K digestion	[41]
Antihypertensive	Wheat gliadin hydrolysate	IAP	2.7 µM	ACE inhibitor; acid protease digestion	[42]
Antihypertensive; antioxidant	Rice protein hydrolysate	VNP, VWP	6.4, 4.5 µM	ACE inhibitor; alcalase and trypsin digestion	[43]
Antihypertensive; antioxidant	Rice bran protein hydrolysate	YSK	76 µM	ACE inhibitor; high DPPH radical scavenging activity; trypsin digestion	[44]
Antihypertensive	Rice bran protein hydrolysate	GSGYF	3.98 µM	ACE inhibitor; pepsin and trypsin digestion	[45]
Antihypertensive	Rice bran vicilin-like protein	LRA, YY	0.25, 0.5 mg/kg	ACE inhibitor; antihypertensive effect in SHRs; thermolysin digestion	[46]
Anticancer	Barley lunasin peptide	SKWQHQQDSCRKQKQGVNLTP CEKHIMEKIQGRGDDDDDDDDD	–	Inhibits histone acetyltransferase and Rb hyperphosphorylation; increases expression of tumour suppressors	[47]
Antihyperglycemic	Wheat gluten	ILDL, ILLPGAQDGL	1121.1, 145.5 µM	DPP-IV inhibitor; alcalase digestion	[48]
Antihyperglycemic	Oat globulin	GDVVALPA, DVVALPAG	–	DPP-IV inhibitor; alcalase and flavourzyme digestion	[49]
Antihyperglycemic	Oat globulin	LQAFEPLR	103.5 µM	DPP-IV inhibitor; alcalase digestion	[50]
Anti-obesity; anti-Alzheimer’s; anticancer	Rice bran protein hydrolysate	EQRPR	–	Insulin-like differentiation of preadipocytes; reduction in cytotoxicity of amyloid-induced neuroblastoma cells; antiproliferative effects on colon, breast, lung and liver cancer cell lines; alcalase digestion	[51,52]
Antioxidant	Finger millet protein	TSSSLNMAVRGGLTR and STTVGLGISMRSASVR	–	DPPH radical scavenging activity; trypsin digestion	[53]
Antioxidant	Sorghum kafirin	YLRQ, AQVAQ, AMCGVV	–	DPPH radical scavenging activity; papain digestion	[12]
Antioxidant; antihypertensive	Corn prolamin	MI/LPP	220 µg/mL (antioxidant), 70.32 µg/mL (ACE)	DPPH radical scavenging activity; ACE inhibitor; alcalase digestion	[54]

**Table 2 molecules-27-03746-t002:** Bioactive peptide sequences from some proteins derived from pulses.

Activity	Type of Protein	Bioactive Peptide Sequence	IC50	Description	Reference
Antihypertensive	Chickpea legumin	MDLA, MDFLI, MFDL	0.01–0.02 mg/mL	ACE inhibitor; alcalase digestion	[55]
Antihypertensive	Mungbean protein isolate	KDYRL, VTPARLR, KLPAGTLF	26.5, 82.4, 13.4 µM	ACE inhibitor; alcalase digestion	[56]
Antihypertensive	Lentil globulin	KLRT, TLHGMV, VNRLM	0.13 mg/mL	ACE inhibitor; α-amylase, pepsin and pancreatin digestion	[57]
Antihypertensive	Pea globulin	GGSGNY, DLKLP, GSSDNR, MRDLK, HNTPSR	0.07 mg/mL	ACE inhibitor; α-amylase, pepsin and pancreatin digestion	[58]
Antihypertensive; antihyperglycemic	Common bean protein hydrolysate	KTYGL, KKSSG	0.09 and 0.20 mg DW/mL (ACE), 0.03 and 0.64 mg DW/mL (DPP-IV)	ACE and DPP-IV inhibitor; pepsin and pancreatin digestion	[59]
Anticancer	Common bean protein hydrolysate	GLTSK, LSGNK, GEGSGA, MPACGSS, MTEEY	-	Antiproliferative effects in HCT116 and RKO cell lines; sequential enzyme digestion	[60]
Anti-inflammatory	Common bean protein hydrolysate	γ-EV	-	Anti-inflammatory activity in intestinal epithelial Caco-2 cells; synthetic peptide	[61]
Antiglycemic; antihypertensive; antioxidant	Lentil protein	SDQENPFIFK, HGDPEER, ATAFGLMK	0.39 mg/mL (ACE)	ACE, α-glycosidase and maltase inhibitor; antioxidant activities; savinase digestion	[62]
Antimicrobial	Lentil defensin peptide	KTCENLSDSFKGPCIPDGNCNKHCKEKEHLLSGRCRDDFRCWCTRNC	-	Inhibits growth of *Aspergillus niger*	[63]
Cholesterol-lowering; anti-inflammatory	Chickpea protein hydrolysate	RQSHFANAQP (CPe-III)	-	Antihyperlipidemic and anti-inflammatory effects in Kunming mice; synthetic peptide	[64]
Cholesterol-lowering	Chickpea protein hydrolysate	VFVRN	-	HMGR inhibitor, decreases TC synthesis; alcalase digestion	[65]
Cholesterol-lowering	Lupin protein	LILPKHSDAD, LTFPGSAED	147.2, 68.4 µM	HMGR inhibitor, synthetic peptides	[66]
Cholesterol-lowering	Lupin protein	LILPHKSDAD	1.6 μM	PCSK9 inhibitor, HMGR inhibitor, synthetic peptide	[67]

**Table 3 molecules-27-03746-t003:** The digestible indispensable amino acid score for cereal grains and pulses. The DIAAS score is determined by the lowest digestible indispensable amino acid value in the protein mixture. SAAs: sulfur amino acids (Met + Cys).

Item	DIAAS Score (%)	Reference
Oats	43–57 (Lys)	[84,85,86]
Dehulled oats	77 (Lys)	[87]
Wheat	43–48 (Lys)	[80,81,87]
Dehulled barley	51–77 (Lys)	[81,87]
Rye	47–56 (Lys)	[81,87,88]
Rice protein concentrate	37 (Lys)	[86,89]
Brown rice (cooked)	42 (Lys)	[79]
Polished white rice	37–64 (Lys)	[79,81,87]
Foxtail millet (cooked)	10–22 (Lys)	[78,79]
Corn	36–48 (Lys)	[79,87]
Sorghum	29–45 (Lys)	[81,90]
Split green peas (cooked)	46 (SAAs)	[91]
Split yellow peas (cooked)	73 (SAAs)	[84]
Chickpeas	83–89 (SAAs)	[82,83]
Peas	58–70 (SAAs)	[80,82,86]
Pea protein concentrate	62–82 (SAAs)	[82,83,86]
Pigeon peas	57 (SAAs)	[83]
Fava beans	55 (SAAs)	[81]
Kidney beans (cooked)	51–58 (SAAs)	[84,86]
Black beans (cooked)	43–49 (SAAs)	[83,84]
Pinto beans (cooked)	60–83 (SAAs)	[84,92]
Navy beans (cooked)	65 (SAAs)	[84]
Mung beans (cooked)	93 (Val)	[78]
Whole green lentils (cooked)	49–58 (SAA)	[84,91]
Split red lentils (cooked)	50–54 (SAA)	[84,91]
Lupins	68 (SAA)	[80]

## Data Availability

The data presented in this study are available on request from the corresponding author.

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
