# Peer review of "Health Benefits of Cereal Grain- and Pulse-Derived Proteins"

_molecules, 2022, doi:10.3390/molecules27123746_

Round 1
Reviewer 1 Report
This is a well-written and interesting manuscript dealing with the health-promoting effects of proteins originated from cereal grains and pulses. This manuscript can be published after some modifications.
-Please choose some keywords which are not present in the title.
-L106: “released by proteolysis”: As you know, there are different methods to release biologically active peptides from proteins; so mention the methods in this part which can improve the quality of your manuscript.
- Recently, the cereals grain and pulse derived proteins are widely used to fabricate carriers for bioactive delivery, but you did not mention this matter in your introduction part.
-There are some review articles and book chapters on this topic (cereals and pulses health benefits in the past (as mentioned following). What is the novelty of your review manuscript compared to these studies? Please justify this matter.
- Liangli, L. Y., Tsao, R., & Shahidi, F. (Eds.). (2012). Cereals and pulses: nutraceutical properties and health benefits. John Wiley & Sons.
- López‐Barrios, L., Gutiérrez‐Uribe, J. A., & Serna‐Saldívar, S. O. (2014). Bioactive peptides and hydrolysates from pulses and their potential use as functional ingredients. Journal of Food Science, 79(3), R273-R283.
- Mudryj, A. N., Yu, N., & Aukema, H. M. (2014). Nutritional and health benefits of pulses. Applied Physiology, Nutrition, and Metabolism, 39(11), 1197-1204.
- Rebello, C. J., Greenway, F. L., & Finley, J. W. (2014). Whole grains and pulses: A comparison of the nutritional and health benefits. Journal of agricultural and food chemistry, 62(29), 7029-7049.
- Tak, Y., Kaur, M., Amarowicz, R., Bhatia, S., & Gautam, C. (2021). Pulse Derived Bioactive Peptides as Novel Nutraceuticals: A Review. International Journal of Peptide Research and Therapeutics, 27(3), 2057-2068.
- One of the most important reasons for the increase in interest in plant proteins is the increase in vegetarian diets, so it is better to mention this issue in your article.
- Is there any study about the effect of these bioactive peptides on the COVID-19?
- What are the challenges for the production and consumption of these bioactive peptides? Can we use them for the production of functional foods? I think you should add a section named “challenges and future trends” to your manuscript to answer these questions.
Author Response
To Whom It May Concern:
We would like to thank you and the reviewers for carefully critiquing our manuscript and for the opportunity to submit a revised version. We appreciate the feedback and have addressed all the points raised by the reviewers. As per your request, revisions/edits made on the revised manuscript have been highlighted. I hope this revised version will be found acceptable for publication in Molecules. Please find attached the response to the reviewers.
Reviewer #1:
-Please choose some keywords which are not present in the title.
New keywords: plant protein; cereal grains; pulses; bioactive peptides; chronic disease; disease prevention
-L106: “released by proteolysis”: As you know, there are different methods to release biologically active peptides from proteins; so mention the methods in this part which can improve the quality of your manuscript.
As per reviewer’s suggestion, we have included additional methods to release bioactive peptides in the revised manuscript (L116).
- Recently, the cereals grain and pulse derived proteins are widely used to fabricate carriers for bioactive delivery, but you did not mention this matter in your introduction part.
Thank you for pointing this out. We have mentioned this briefly in the revised manuscript (L112).
-There are some review articles and book chapters on this topic (cereals and pulses health benefits in the past (as mentioned following). What is the novelty of your review manuscript compared to these studies? Please justify this matter.
- Liangli, L. Y., Tsao, R., & Shahidi, F. (Eds.). (2012). Cereals and pulses: nutraceutical properties and health benefits. John Wiley & Sons.
- López‐Barrios, L., Gutiérrez‐Uribe, J. A., & Serna‐Saldívar, S. O. (2014). Bioactive peptides and hydrolysates from pulses and their potential use as functional ingredients. Journal of Food Science, 79(3), R273-R283.
- Mudryj, A. N., Yu, N., & Aukema, H. M. (2014). Nutritional and health benefits of pulses. Applied Physiology, Nutrition, and Metabolism, 39(11), 1197-1204.
- Rebello, C. J., Greenway, F. L., & Finley, J. W. (2014). Whole grains and pulses: A comparison of the nutritional and health benefits. Journal of agricultural and food chemistry, 62(29), 7029-7049.
- Tak, Y., Kaur, M., Amarowicz, R., Bhatia, S., & Gautam, C. (2021). Pulse Derived Bioactive Peptides as Novel Nutraceuticals: A Review. International Journal of Peptide Research and Therapeutics, 27(3), 2057-2068.
Thank you for providing us with the list of references. We have now added a new section on the novelty of our review in the revised manuscript (Starting at L129)
- One of the most important reasons for the increase in interest in plant proteins is the increase in vegetarian diets, so it is better to mention this issue in your article.
This has been included in the revised manuscript (L109).
- Is there any study about the effect of these bioactive peptides on the COVID-19?
The number of research article on plant proteins/peptides and COVID 19 are very limited (most research focuses on milk peptides). After careful review, we decided to remove the lines related to COVID in the cancer section as this is outside of the scope of this review paper.
- What are the challenges for the production and consumption of these bioactive peptides? Can we use them for the production of functional foods? I think you should add a section named “challenges and future trends” to your manuscript to answer these questions.
Thank you for the suggestion, We have created a section briefly summarizing challenges and trends for the production of bioactive peptides (section 2.1)
Reviewer #2:
1.Figure 1: Put the abbreviations in the figure, and modify the layout to facilitate reading.
Thank you for the suggestion. As per reviewer’s suggestion, the figure has been modified and font size has been increased to facilitate reading.
2.The subtitle is not related to the figure 1. Please revise the relationship between the relevant activities in figure 1 and this article.
Thank you for pointing this out. The figure has been moved to a different section and the title of the figure has been modified.
Reviewer #3:
- Section 2. The nutritional complementarity of cereals and pulses in products formulated with both should be discussed somewhere in this section (doi: 10.3390/plants10101999, 10.1007/s13197-017-2537-4 , 10.3390/foods8060199).
This has been discussed in the revised manuscript (Starting at L183)
- Tables 1-2 should be reformatted, by substituting the current reference column with a numbered column (numbered complete references should be placed as a footnote).
As per the reviewer’s suggestion, the tables have been reformatted so that the numbered references in the tables correspond with the bibliography. However, we were not able to include these references with the footnote due to the large number of references and potential interference with the endnote software.
- Table 3: All PDCAAS values must be expressed as ranges (supported with more references) as there are sufficient references of other extreme values for each source/limiting amino acid (e.g.). The authors should check the reported values again as some are wrong (e.g. dehulled Barley is 51 and not 77; doi: 10.1017/S0007114513004273)
Please note that we included DIAAS values, not PDCAAS. However, as per reviewer’s suggestion, more references have been added to supplement the table.
- Section 3. Each preventive action described and commented must be accompanied by a summary table (fed with the same references used) arranged as follows (left to right): Protein source, amount evaluated, protein bioactive (e.g. amino acid, a bioactive peptide, etc.), proposed mechanism, reference (sequentially numbered and its description as a footnote).
In case the systematization of the information so indicates, an additional figure could be included on the source-specific benefits, which helps to generate personalized nutritional recommendations (precision medicine).
Thank you for the suggestion. After careful review of the text, we decided not create another table as it would look very similar to the existing bioactive peptide table. However, as per the reviewer’s suggestion, a new figure (Fig: 2) has been implemented into the hypercholesterolemia section.

Reviewer 2 Report
This review article uses cereal Grain and Pulse derived proteins to prevent diseases and risk factors such as cardiovascular disease, hyperglycemia and diabetes. The content of the article is complete, and there are only two suggestions as follows:
1.Figure 1: Put the abbreviations in the figure, and modify the layout to facilitate reading.
2.The subtitle is not related to the figure 1. Please revise the relationship between the relevant activities in figure 1 and this article.
Author Response

(The authors gave the same response as above.)

Reviewer 3 Report
The authors present a quasi-systematic review on the nutritional/functional value of cereal grains and pulses, focusing its discourse on the proteinic fraction (7-15 and 18-30g.100g-1, respectively) of these edible sources. The authors organized the information and discourse, first addressing the differences in protein composition (Sections 1-2: Total protein-amino acid profile-bioactive peptides; 2 systematic tables, 1 figure) from specific sources to later justify, in a narrative way, the evidence on the role of the consumption of these specific protein sources in the prevention and control of various physiological deviations and diseases (Section 3; no figures nor tables).
- Section 2. The nutritional complementarity of cereals and pulses in products formulated with both should be discussed somewhere in this section (doi: 10.3390/plants10101999, 10.1007/s13197-017-2537-4 , 10.3390/foods8060199).
- Tables 1-2 should be reformatted, by substituting the current reference column with a numbered column (numbered complete references should be placed as a footnote).
- Table 3: All PDCAAS values must be expressed as ranges (supported with more references) as there are sufficient references of other extreme values for each source/limiting amino acid (e.g.). The authors should check the reported values again as some are wrong (e.g. dehulled Barley is 51 and not 77; doi: 10.1017/S0007114513004273)
- Section 3. Each preventive action described and commented must be accompanied by a summary table (fed with the same references used) arranged as follows (left to right): Protein source, amount evaluated, protein bioactive (e.g. amino acid, a bioactive peptide, etc.), proposed mechanism, reference (sequentially numbered and its description as a footnote).
- In case the systematization of the information so indicates, an additional figure could be included on the source-specific benefits, which helps to generate personalized nutritional recommendations (precision medicine).
Author Response

(The authors gave the same response as above.)

Round 2
Reviewer 3 Report
Thank you very much for considering most of my recommendations
Author Response
Thank you!